# Collaboration and Task Planning of Turtle-Inspired Multiple Amphibious Spherical Robots

**DOI:** 10.3390/mi11010071

**Published:** 2020-01-09

**Authors:** Liang Zheng, Shuxiang Guo, Yan Piao, Shuoxin Gu, Ruochen An

**Affiliations:** 1Graduate School of Engineering, Kagawa University, Takamatsu, Kagawa 761-0396, Japan; zheng_cust@163.com (L.Z.); s19d501@stu.kagawa-u.ac.jp (R.A.); 2Jilin Agricultural Science and Technology University, Jilin 132101, China; 3Changchun University of Science and Technology, Changchun 130022, China; 4Department of Intelligent Mechanical Systems Engineering, Kagawa University, Takamatsu, Kagawa 761-0396, Japan; 5Key Laboratory of Convergence Medical Engineering and System and Healthcare Technology, the Ministry of Industry Information Technology, School of Life Science, Beijing Institute of Technology, Beijing 100081, China; 6School of Control Engineering, Chengdu University of Information Technology, Chengdu 610225, China; gsx@cuit.edu.cn

**Keywords:** amphibious spherical robot, collaborate robot, rigid formation, underwater communication, follower robot, underwater robot

## Abstract

Amphibious Spherical Robots (ASRs) use an electric field to communicate and collaborate effectively in a turbid water of confined spaces where other mode communication modalities failed. This paper proposes an embedded architecture formation strategy for a group of turtle-inspired amphibious robots to maintain a long distance-parameterized path based on dynamic visual servoing. Inspired by this biological phenomenon, we design an artificial multi-robot cooperative mode and explore an electronic communication and collaborate devices, the control method is based in particular on underwater environment and also conduct a detailed analysis of control motion module. The objectives of control strategies are divided into four categories: The first strategy is that the leader robot controls the action of the overall robots to maintain collaborate together during motion along a desired geometric path and to follow a timing law that the communication efficiency and the arrival times to assigned sites. Furthermore, we design an adaptive visual servoing controller for trajectory tracking task, taking into account system dynamics with environment interactions. After that, the third strategy is a centralized optimization algorithm for the redistribution of target mission changes. Finally, this paper also proposes a new method of control strategies in order to guarantee that each robot in the team moves together according to the preset target toward its location in the group formation based on communication and stability modules.

## 1. Introduction

With the increasing extensive requirements of high maneuverability, long duration, energy-saving and even stealth for Autonomous Underwater Vehicle (AUV), many kinds of bio-inspired amphibious robots based on ideas of biological systems, such as fishes, snakes, crabs, whales, turtles and other reference animals [1,2,3,4,5,6,7,8], which have showed better characteristics for adapting to underwater, terrestrial and aerial environments. In recent years, there has been a growing interest in using large numbers of the amphibious spherical robot to realize communication and cooperation [9].

This study has mainly focused on dynamic performance, locomotion control, and swimming efficiency to improve the flexibility and stability of amphibious spherical robots. At the same cost and benefit, such group robots can carry more sensors to collect more needed data, improving adaptability and flexibility during task execution, such as patrolling seaports, search and rescuing [10,11]. On basis of ordinary motion behavior, formation initialization and movement planning are also studied in this paper. An auction-based on initialize formation planning method with time-optimal route plan is proposed, usable the formation problem has been transformed to task allocation problem, that is, the ASRs in follower roles bid for the position when the robot in the queue is damaged or detached from the formation. The robot in leader role perform centralized planning and distributed biddings can accomplish the mission of position allocation while reducing the complexity of computation dimension. A study on multiple amphibious spherical robots for collaboration and task planning based on dynamic visual servoing is presented, we introduce related research on various approaches in collaboration and task assignment control strategies. Garcia-Fidalgo E, Ortiz A et al. developed a vision-based control algorithm for an AUV in a multi-robot undersea intervention task used a visual servoing module generates the required velocities for controlling the platform according to the estimated position of the target in the image plane [12,13,14]. Distributed cooperative control strategies can realized, including consensus [15,16], flocking [17,18], formation [19,20], containment [21,22], etc. Pearce et al. proposed an optimization framework that generates task assignments and schedules for a human-robot team to improve both time and ergonomics and demonstrate its use in six real-world manufacturing processes that are currently performed manually [23]. Other planning approaches include rule-based [24] or a Satisfiability Modulo Theories (SMT)-based algorithm [25]. Based on the adjacent output measurement, a distributed observer type containment protocol is proposed for linear multi-leader and multi-agent systems in [26].

In addition, in the study of multi-robot collaborative control, finite-time control is also used to improve computational efficiency. In [27], a novel decentralized output sliding-mode Fault-Tolerant Control (FTC) design for heterogeneous Multi-Agent Systems (MASs) with matched disturbances, unmatched nonlinear interactions, and actuator faults. Reference [28] addresses distributed finite-time attitude control of multiple rigid bodies when multiple stationary or dynamic leaders occur. One of the essential capacities of robots, communication and collaboration is particularly challenging for bio-inspired underwater robots that typically work in a limited water environment, stringent power and size constraints. Widely more popular communication systems method (e.g., 3G/4G, Bluetooth) that are commonly used in the air, can barely work underwater. Hotspot optical communication cannot work in unclear water due to the requirement for a line of sight while communicating. Acoustic communication has the problem of large doppler shifts and multi-path effects when robots are engaged in highly confined aquatic environments (e.g., shallow water, narrow pipes, tunnels, and caves), therefore, an alternative communication modality would be significant and in urgent need of being developed for the small underwater robot. Zhang’s deep learning based on communications method and with a considerable reduction in training overhead compared to the traditional communication mode [29]. Park et al. designed a channel models and estimation, digital signal processing to improve underwater communication efficiency [30]. In particular, a time diversity Passive Time Reversal (PTR) system is proposed and it uses the time diversity of the channel to compensate for the channel variations to improving communication quality [31]. Therefore, the problem of underwater communication efficiency is the theoretical basis for the realization of the multi-robot cooperation and collaboration. Centelles D et al. proposed a UnderWater robot Simulator-NET (UWSim-NET) modeling can minimize the impact of their limited performance in real-time robotic interventions [32].

Base on the above literature review, one sees that there also exist some aspects that need to be continuously optimized. This paper proposed a new type amphibious robot contained communicate and collaborate modules and designed a novel formation control method, which means the problem of controlling the relative position and orientation in a group according to some desired pattern for executing a given task. Different approaches have been proposed to tackle the robot formation control problem, behavior-based, follow-structure, and leader-follower approaches. The formation control is prevalent in the group activities of fish, birds and other animals. To a specific geometric shape or maintain a relatively constant positional relationship during the movement so that they can unite to find, hunt or resist predators more effectively. Naturally, inspired by tortoises, formation behavior should also be applied to the collaborative exploration process of multiple robots in an unknown environment. The introduction of the method will inevitably increase the efficiency of collaborative exploration and enhance the robustness of the system. Therefore, the formation of control research has gradually become an important content in multi-robot systems, especially in military surveillance, aerospace, industrial and agricultural production. As shown in Figure 1, we describe the successful application of the proposed method to complete multi-robot communication and collaboration that its detection accuracy is similar to or higher than that of other related method with shorter processing time. This paper also proposes a distance-based optimal planning method based on a follow positioning mechanism. This method converts the multi-robot formation problem into a task assignment problem. The follower robot used the following algorithms, while the leader robot performs centralized planning according to the algorithms. The combination of the optimization methods reduces the centralized planning time to calculate the dimension and completes the task of assigning the position of the formation control system. In addition, using the real-time monitoring mechanism generated the algorithms, it is possible to re-plan the formation at the time of task failure or mission demand and maintain the integrity of the formation control. Physics experiments verify the feasibility of the time-optimized and real-time multi-robot planning algorithm based on the control algorithm. Compared with other formation planning algorithms, the proposed formation distance optimal strategy has the advantages of a short time and high efficiency in formation control processing. In many situations, the problem of multi-robot formation control might be reduced to the tracking problem, controller for trajectory tracking have been presented for various applications of the ASRs. This paper developed with the control problem of groups on nonholonomic ASRs and the convoy-like formation and proposes new control strategies for ensuring that each robot of the group drives autonomously toward its location and then steadily maintains its relative position in the group to complete or capture the targets.

ASRs needs a complete vision acquisition device to perform complex underwater tasks, this paper demonstrates the first try on uncalibrated visual tracking control of the amphibious spherical robot in an underwater environment. Fan Xu et al. proposed a controllability of an octopus tentacle-like soft robot arm operating in an underwater environment by visual servoing controller [33]. Jiyong Li et al. presents an uncalibrated visual servoing scheme for Underwater Vehicle Manipulator Systems (UVMSs) with an eye-in-hand camera under uncertainties [34]. A tracking method adopting the cloud-like model data association algorithm is presented in order to track underwater multiple targets in [35]. We propose an adaptive control algorithm to solve the uncalibrated problem and to compensate for distance error between dynamic target and robot. Using the visual servoing and Proportion Integral Differential (PID) control method, the rotational speeds of the two drive motors are adjusted in real-time according to the different distances measuring of the target. If the ASR is close to the target, the visual servoing motor will stop working and return to the mode of continuing to find the next target. To our knowledge, there is no research on dynamic visual servoing for amphibious spherical robots.

The rest of this paper is organized as follows. Section 2, we introduced the mechanical design and evaluation of the module performance. In Section 3, we presented and completed the modeling of motion control for three ASRs. In Section 4, we did the most important related underwater experiments about communication and collaboration modules with formation control strategies. Finally, conclusions and future work are presented in Section 5.

## 2. Design Criteria

### 2.1. Mechanical Design

In this paper, we designed a novel ASR, which can move on land and underwater to perform complicated tasks. The robot has two actuating modes: quadruped walking mode and water-jet propulsion mode. the structure and dimension of the amphibious spherical are moderate. The amphibious spherical robot also has the ability of movement from the ground to underwater. The best designation of the amphibious robot can alter drive mode between water-jet propulsion mode and quadruped walking mode with the structure of complicated propulsion mechanisms. The amphibious robot has a capacity of movement from the ground to underwater [36,37].

The structure of amphibious robot composed of two parts that contained a sealed transparent upper hemisphere and have a transparent quarter spherical shells, the shells can be opened and closed through the main circuit board sends a digital signal to control a servoing motor [38,39]. Between the upper hemisphere and lower hemisphere, a plastic plate for carrying the circuit boards, four actuating legs units and the battery installed on this plate. The communication module must be installed at the bottom so that it can be in contact with water.

In past research in our lab, our team designed a ASR that can move from the water to the ground without manpower and vice versa. Four legs robot always have a good performance on even ground. As shown in Figure 2, the robot has four actuating legs units, each leg constituted of two servoing motors and a water-jet motor, they were symmetrically installed under the lower hull [35,36]. The control circuits system, sensors, and power supply are installed in the sealed upper hemisphere, it has waterproof performance. When the robot worked in underwater mode, the lower hulls closed, and four water jet motors in the actuating units provided vectored thrust through strip-type holes in the lower hulls to release motion with 6 Degrees of Freedoms (DoFs). When the robot was in land mode, the lower hulls can be opened up and the actuated units stretched out to walk quadrupedally under the driving force of eight servoing motors [40,41,42]. More importantly, this paper proposed a novel generation of the ASR with communication and visual servoing modules can complete the complicated movement and multi-robot cooperation and capture the target no matter what robot moving on land and underwater.

### 2.2. Communication Module

Developing a novel artificial communication and cooperation multi-robot is difficult due to the unexplored scheme and implementation method. For example, what kind of mode should the electric field be emitted and measured by? What kind of form (analog and digital) should the signal transfer by? Further, it is more challenging to design such an artificial communication and cooperation system for underwater robots which usually volume constraint, stringent power, limited hardware and conventional mobility in robotics. This section addresses these issues in the design of communication and cooperation system, the electromagnetic actuation system is shown in Figure 3.

First, due to the limitation of the volume, the main controller selects an advanced Reduced Instruction Set Computing (RISC) machines as the main controller, which can increase the real-time characteristic of the control system, we choose to emit and measure the electric signal in the form of voltage because voltage measurement is a common solution to the traditional communication system. Second, we choose a digital method for communication and cooperation since it is capable of higher bit rate delivery, higher anti-interference and easier integration with the robot microcontroller compared with the analog method. Third, this designation also used the Universal Asynchronous Receiver/Transmitter (UART) protocol to transfer information with a microcontroller, it can provide a convenient way to tune communication parameters (e.g., baud rate, data length and check bit) in practice. Moreover, UART also simplifies the demodulation circuit in system design [39]. Finally, the system scheme is formed and contains a transmitting unit, a receiving unit and two pairs of the control unit.

To acquire a better performance, the transmitting system should impose an electric field as strong as possible while the receiving system should be able to sense the electric signal as weak as possible. However, if the system can impose an extremely strong electric field as well as pick up very weak electric signal, its circuit complexity and power consumption will be unaffordable to ASRs, considering the limitations like the small size of upper hemisphere, stringent power and low computational capability of underwater robots, the communication system is particularly designed to be low power, small size and low complexity. The Micron Data Model (MDM) acoustic communication sensor meets the above requirements, which provides a serial data passing to the terminal, the MDM has an internal buffer of 256 bytes. The incoming serial data rate must take into account the acoustic transmission rate of 40 bit/s, the acoustic radiation pattern is approximately omni directional and the MDM will operate in both lateral and longitudinal posture.

### 2.3. Visual Servoing Evaluation

For the special and variable condition of the underwater environment, there are three necessary algorithms used to the underwater robot motion control. The first is a controller model based on a Linear Quadratic Regulator (LQR), and the second is a motion equation of a control system based on nonlinear state Feedback Linearization (FL). Finally, the most commonly used in most research is PID algorithm. In the next process, this paper compares and analyzes the principles of this three control algorithms. The LQR regulator solves the control problem by minimizing performance indicators, the dynamic equation is as follows:
(1)$$J=\frac{1}{2}\int_{0}^{\infty }\left[{\left(x-x_{R}\right)}^{T}Q(x-x_{R})+{\left(u-u_{R}\right)}^{T}R(u-u_{R})\right]d_{t}$$
where x is the state vector and $x_{R}$ is the reference state vector. u is the input vector and $u_{R}$ is the input reference vector. Q and R is the transition matrix of states and inputs, respectively. As shown in Figure 4, the control system consists of two nested closed-loop controls. The inner loop is speed control and the outer loop is position control. Two controls coefficient is $k_{0}$ and $k_{1}$ are two independent gain variables for the LQ control. The state gain of the LQ controller according to Equation (1) is a diagonal matrix, so each term on the diagonal is normalized to the matrix Q, as shown in Equation (2):
(2)$$q_{i}={\left(\frac{1}{\max error_{i}}\right)}^{2}$$

By adjusting the parameter *R* of the dynamic system, the parameter relationship between the response time and the required control force is selected, and the value of the parameter *R* is gradually adjusted in the actual verification. The basic method of the nonlinear state feedback control is to transform the nonlinear dynamic model into a set of independent integration chains through an algorithm, as shown Equations (3) and (4) below:
(3)$$\overset{\cdot \cdot }{\eta }=\lambda^{n}$$
(4)$$\overset{\cdot }{v}=\lambda^{b}$$
where η denotes the position and orientation vector in the coordinate system. v denotes the linear velocity. $\lambda^{n}$ and $\lambda^{b}$ are acceleration and acceleration vector. In the controller schematic, some valid data can obtained through sensors.The controller closed-loop schematic is shown in Figure 5.

One of the most important and practical controllers has been widely known as the PID controller. The overall formula of the PID controller can be given as:
(5)$$u_{PID}(t)=K_{p}e(t)+K_{1}\int_{0}^{t}e(\tau )+K_{D}\frac{de(t)}{dt}$$
where $K_{p}$, $K_{1}$ and $K_{D}$ are defined as proportional, integral and differential parts, respectively. The error between the desired value and the true value of this system is given by e, the control principle is shown in Figure 6.

This paper uses the PID controller because the PID control algorithm is a traditional control method compared with LQR, FL, etc. The advantage of the PID control is that the parameters are relatively easy to adjust, which is suitable for the non-linear motion mode of the underwater robot. It can be simplified into a basically linear system by simplifying it. The camera sensor is mounted on the circuit board which installed on the middle of the spherical robot. Employing the Pulse Width Modulation (PWM) port to control simultaneously two sets of servoing motors, so that the direction of motion can be controlled according to the captured target location. Through the camera detection algorithm, the robot adopts the camera sensor to capture the image position of the largest color block and calculate the deviation of image center, and then draw a cross at the center and a rectangle around the periphery of the deviation. After that, the PID algorithm is used to calculate the parameters of the PWM speed of the left and right motors, the calculated parameters are sent to the motor control circuit board, the robot will advance move to the target which captured objective by the camera.

In order to verify the real-time characteristic of the camera sensor to detect moving object and to control the accuracy of the motor thrust. It is necessary to design an experimental platform that can realize a camera to control two motors at the same time to achieve the purpose of capturing a moving target. The experimental platform is shown in Figure 7, the experimental objective is to control the rotation of the motors by detecting the target through camera sensor. For the purpose of verifying the rotational power of the motors, the experimental design of two motors rotating in a water tank filled with water, so that the rotational speed and motor direction are more clearly observed. The snapshot sequence in Figure 8 is extracted from a video recorded live without any modification. The experiment is mainly to determine two modes, the first and second mode is to find the target by two drive motors to rotate at a fixed frame and keep the low speed to find the target, as shown in Figure 8a,b. The third is tracking mode, in this mode, if the camera detects the target, the two motors will rush to the target at a fast speed, as shown in Figure 8c.

## 3. Motion Algorithm and Modeling

### 3.1. Formation Control Modeling

This section is devoted to the control problem of three ASRs that two robots can follow a leader robot together form a desired rigid formation. The concept of formation is closely related to leader-follower formation (LFF), under the LFF mode, the leader robot moves along a predefined trajectory, while the follower robots maintain a desired distance and orientation to the leader robot. In this section, we present an approach for tackling the rigid formation control problem by the stochastic motion transformation. The proposed control strategy allows the robot to execute formation maneuvers, departing from the formation, splitting the formation and merging into the formation. In our setting previously, the control algorithm dominates the rule of the formation.

Figure 9 shows the coordination robot procession is composed of the leader robot and two follower robots. The coordinates of follower robots are fixed corner position and provides a coordinate reference for the leader robot and another follower robot. Location error is described by the positional relationship between the leader robot and follower robots. We assume that the selected reference trajectory ensures that the trajectories of formation robots satisfied the system physical constraints, $q_{l}=\left(x_{l},y_{l},\theta_{l}\right)$ is an inertial coordinate. The equations are regarded as the kinematic equation, the modeling of the leader robot is given by:(6)$$q_{l}=\left[\begin{array}{l} \overset{\cdot }{x_{l}}\\ \overset{\cdot }{y_{l}}\\ \overset{\cdot }{\theta_{l}} \end{array}\right]=\left(\begin{array}{cc} \cos\theta_{l} & 0\\ \begin{array}{l} \sin\theta_{l}\\ 0 \end{array} & \begin{array}{l} 0\\ 1 \end{array} \end{array}\right)\left(\begin{array}{l} v_{l}\\ \varpi_{l} \end{array}\right)=\left[\begin{array}{l} v_{l}\cos\theta_{l}\\ v_{l}\sin\theta_{l}\\ \varpi_{l} \end{array}\right]$$
where ($x_{l}$, $y_{l}$) is the position coordinate of the centroid of the leader robot, $\theta_{l}$ is the attitude angle, and $v_{l}$ is the velocity of the follower 1 robot. $\omega_{l}$ is the angular velocity of the follower 1 robot. So the coordinate equation of two robots is:(7)$$\left\{ \begin{array}{l} x_{v}=x_{l}-L_{lv}\cos(\theta_{v}-\phi_{vv})\\ y_{v}=y_{l}-L_{lv}\sin(\theta_{v}-\phi_{vv})\\ \theta_{v}=\theta_{f} \end{array}\right.$$
(8)$$\left\{ \begin{array}{l} x_{v}^{\prime }=x_{f}+L_{fv}\cos\theta_{\begin{array}{l} f \end{array}}\\ y_{v}^{\prime }=y_{f}+L_{fv}\sin\theta_{\begin{array}{l} f \end{array}}\\ \theta_{v}^{\prime }=\theta_{f} \end{array}\right.$$
where ($x_{v}$, $y_{v}$) is the position of the centroid of the leader robot, ($x_{v}^{\prime }$, $y_{v}^{\prime }$) is the position of the centroid of the follower 1 robot, $\theta_{v}$ is the attitude angle of the leader robot, $\theta_{f}$ is the attitude angle of the follower 1 robot, $\phi_{vv}$ is a deflection angle of the follower 1 robot, we can establish contact equations between two follower robots, the mathematical model of the tracking error can be derived by:(9)$$\left\{ \begin{array}{l} e_{FT}^{x}=x_{v}-x_{v}^{\prime }=\left[x_{l}-L_{lv}\cos(\theta_{v}-\phi_{vv})\right]-x_{f}-L_{fv}\cos\theta_{f}\\ e_{FT}^{x}=y_{v}{-y}_{v}^{\prime }=\left[y_{l}-L_{lv}\cos(\theta_{v}-\phi_{vv})\right]-x_{f}-L_{fv}\cos\theta_{f}\\ e_{FT}^{\theta }=\theta_{f}-\theta_{l} \end{array}\right.$$


On the Leader-to-Formation Stability (LFS), the concept of LFS has been thoroughly used in formation control. The LFS puts emphasis on the leader robot motion and an initial affects the formation interconnection errors. Therefore, relative position error will yield during the movement, the error dynamics equation given by:
(10)$$e_{FT}=\left[\begin{array}{l} e^{x}{}_{FT}\\ e^{y}{}_{FT}\\ e^{\theta }{}_{FT} \end{array}\right]=\left[\begin{array}{l} \Delta x-L_{lv}\cos\Delta \theta_{lv}-L_{fv}\cos\Delta \theta_{f}\\ \Delta y-L_{lv}\sin\Delta \theta_{lv}-L_{fv}\sin\Delta \theta_{f}\\ \Delta \theta_{fl} \end{array}\right]$$
where ΔX, ΔY devote to the lateral and longitudinal distance of the leader robot and follower robot, $\Delta \theta_{lv}$ is the lateral angle, $\Delta \theta_{f}$ is the longitudinal angle. The ultimate goal of behavior control is to bring errors closer to 0, so we can conclude Equation (11):(11)$$\underset{t\to \infty }{\lim}e_{F_{i}T_{j}}(t)=0,i,j\in [2,n]$$

Since the robot cannot move from time 0 to t during the movement and the moving speed cannot equal to 0, which means that the velocity of the formation robot does not have a linear equation starting from 0, so the equation of Equation (10) is rewritten as:(12)$$e_{F_{i}T_{j}}(t)\leq \varepsilon ,i,j\in [2,n]$$

From formula of Equation (12) above, where the $\varepsilon ={\left[\begin{array}{ccc} \varepsilon_{l} & \varepsilon_{l} & \varepsilon_{\theta } \end{array}\right]}^{T}$, $\varepsilon_{l}$ indicates the positional error in the x-axis and y-axis direction, $\varepsilon_{\theta }$ is the constant of attitude angle error. If the distance error value between follower robots meets $\left|e_{F_{i}T_{j}}^{x}(t)\right|<\varepsilon_{l},\left|e_{F_{i}T_{j}}^{y}(t)\right|<\varepsilon_{l}$ and error angle value meets $\left|e_{F_{i}T_{j}}^{\theta }(t)\right|<\varepsilon_{\theta }$, satisfying the above conditions and enters the posture of adjustment motion, so the trigger conditions can be derived by:(13)$$\left|e^{x}{}_{F_{i}T_{j}}{\left(t\right)}^{}\right|<\varepsilon_{l},\left|e^{y}{}_{F_{i}T_{j}}{\left(t\right)}^{}\right|<\varepsilon_{l},\left|e^{\theta }{}_{F_{i}T_{j}}{\left(t\right)}^{}\right|<\varepsilon_{\theta }$$
where the $t\to F_{i}T_{j}PA$ is the time required for posture adjustment, after the posture adjustment, the follower robots will move towards the leader robot, so this behavior is called the behavior toward the target, it is drawn as $\left|e_{F_{i}T_{j}}^{y}(t)\right|<\varepsilon_{l}$. The trigger condition for this behavior is deduced as below equation.
(14)$$\underset{t\to F_{i}T_{j}FA}{\lim}\left|\left|e^{x_{}}{}_{F_{i}T_{j}}(t)\right|<\varepsilon_{\theta }\right|,\left|e^{y}{}_{F_{i}T_{j}}(t)\right|<\varepsilon_{l},i,j\in [2,n]$$
where the $t\to F_{i}T_{j}PA$ is the time required for the forward goal that can be achieved by the actual time for the forward goal.

### 3.2. Bidding-Based Queue

To improve the probability of multi formation control, the shortest and most efficient formation method becomes crucial particularly. In the process of multi-robot performing a task, if the leader robot is damaged or unable to perform the task, the formation model based on the bidding strategies can enable the follower robot to imitate the function of the leader robot, thereby realizing the rearrangement of the queue and achieving the task.

The real-time monitor mechanism raised at the bidding moment can assist the leader robot to plan the formation method again as soon as the leader robot breaks down or task requires, this strategy can maintain the completeness of the formation. Simulation and experiments in formation control are taken to illustrate the feasibility of the proposed auction-based time-optimal strategy (Section 4). The formation distance-optimal strategy is compared with single robot time strategy, the bidding-based approach can assist robots realize a decision-making system similar to human behavior and builds a cooperative allocation strategy. The specific process is shown in Figure 10. The bidding-based process has four main processes: task transfer, bidding, authorization and confirmation. When the leader robot issues a task to the follower robots performing the mission, the two follower robots return a confirmation signal through the sonar module in order to confirm the bid, and the leader robot judges the distance of other robots based on the delay time of the communication module. Relative position can be confirmed that one of the follower robot can bid for request successfully and send a confirmation signal to the follower robot, so that the leader robot re-centers the entire of the queue. The formation process is shown in Figure 11.

## 4. Performance Verification

### 4.1. Experiment I: Robot Stability Evaluation

The experiments analysis under the environment of a square water tank that is 2500 mm in length, 2000 mm in width and 800 mm in height, the water depth is 700 mm. The movement distance is 2500 mm, as shown in Figure 12. Under the control of the stability module, the purpose of this experiment is that robots move about 2500 mm during the time of 0–15 s, the stability control system is installed in the middle of the plate. During the underwater movement, due to the different force of the water flow, the robot will deflect and track the position to the target, stability control module will intervene, the robot returns to the position of the terminal point after deviating from a certain displacement and reaches a state of stable balance.

In Figure 13, when the robot deviates from the preset path by more than 15 degrees, it will trigger a stable closed-loop control system and control the corresponding water-jet motor according to the yaw angle to adjust the movement direction. Figure 14 illustrates the robot moves to target with deviation error curve in y-axis direction. After the intervention of the stability control system, the robot goes back to the center and continues to move with straight line trajectory. In this experiment, the robot receives occasionally the effect of water waves and it will be yield deflect angle, which reduces the effectiveness for the control of the stability system. Therefore, in the program, when the y-axis angle exceeds 15 degrees, the stability control system adjusts the corresponding motor to adjust the robot direction, the moving position can be adjusted, which can allow the spherical robot to continue to move along a preset trajectory. Through this control method, the fixed trajectory motion of a single spherical robot is achieved. It can lay a technical foundation for multi-robot to realize cooperation and collaboration.

### 4.2. Experiment II: Formation Control Evaluation

The experiment is divided into two parts: the first part is that the three ASRs going forward to the target with longitudinal queues. The second part is that the three ASRs going forward to the target with lateral queues. The basic principle is that the robot can realize moving through the stability control system. The gyro sensor controls the x-axis, y-axis and z-axis deflection angle based on the judgment program controls the linear direction. The method of measuring the distance is to differently predict the fixed spacing by the different delay times of the communication module at different distances. This method has the advantage of controlling the real-time detection spacing at different angles and directions. When the leader robot starts to move, the communication module starts to send distance data to the follower robots, the follower robot calculates the delay time by the processor through the serial port. The four control directions of the motor under the control of the gyro sensor and tack the leader robot to move straight and update the position, then feedback the distance data to the main controller.

The total distance length of the three ASRs movements is 2500 mm, and robots realized longitudinal queues following movement under the leadership of the leader robot, as shown in Figure 15, the follower robot first performs the positioning of the leader robot and makes three robots in a fixed position of the similar straight line. In order to more accurately describe the relative position of three ASRs, the experiment collected five equidistant points for detecting the relative distance of the leader robot and the follower 1 robot ($L_{lv}$), the leader robot and the follower 2 robot (L) and the two follower robots ($L_{fv}$) to verify the collaborative control algorithm, as shown in Figure 16. The control principle is defined in the Section 3, the starting and terminal coordinates change of the formation robot are shown in Figure 17, we set the initial coordinate values of leader robot and two follower robots is $q_{l}=\left(625,400,30^{{}^{\circ}}\right)$, $q_{f}=\left(125,400,90^{{}^{\circ}}\right)$ and $q_{v}=\left(265,400,60^{{}^{\circ}}\right)$. The maximum offset position of the *L* is 660 mm, which occurs when the robot moves to 1400 mm and the deviation value is 35 mm. The maximum deviation of $L_{fv}$ occurs at 2300 mm, the distance reaching 295 mm, and the deviation value is 30 mm, as shown in Figure 18. The relative position of $L_{fv}$ is slight and can be ignored. The coordinates of the terminal point are $q_{l}=\left(2300,325,8^{{}^{\circ}}\right)$, $q_{v}=\left(2070,370,20^{{}^{\circ}}\right)$, and $q_{f}=\left(1780,400,25^{{}^{\circ}}\right)$.

Figure 19 shows that the three ASRs realize the cooperative motion with lateral queues and the robot can reach the target under the positioning of the communication system. Figure 20 shows the relative distance positions of the three robots, the initial coordinate values of leader robot and two follower robots is $q_{l}=\left(600,520,60^{{}^{\circ}}\right)$, $q_{f}=\left(125,260,90^{{}^{\circ}}\right)$ and $q_{v}=\left(125,520,60^{{}^{\circ}}\right)$, the terminal coordinates is $q_{l}=\left(2270,500,25^{{}^{\circ}}\right)$, $q_{f}=\left(1900,190,12^{{}^{\circ}}\right)$ and $q_{v}=\left(2190,710,40^{{}^{\circ}}\right)$, as shown in Figure 21. In Figure 22, the L and $L_{fv}$ have a large deviation from the initial position to the target position, the relative distances are 41 mm and 37 mm, respectively, and the deviation of $L_{lv}$ occurs at 1400 mm, reaching 30 mm. Under the influences of water waves, the relative displacement of $L_{fv}$ has undergone a serious change, reaching 90 mm. Under the action of the stability control system, the cooperative task is almost completed. The program sets the leader robot to perform distance positioning calibration every 0.1 s. To avoid excessive distance error in the process of the following cooperation, according to the experiment results, the error curves of the x-axis, y-axis and declination can be obtained, as shown in Figure 23. 

### 4.3. Experiment III: Bidding Strategy for Multi Robot

After the completion of multi-robot collaboration experiments, if the leader robot is damaged or loses the dominant control position, the leader robot and the follower robots need to be changed in the queue, so that maintain the formation sequence to complete the mission objectives.

This experiment is based on the algorithm of robot cooperation, combined with the communication and stability control module, using the method of sending bidding strategy to make follower robots and the leader robot complete the position switching. The follower 1 robot increases the speed of its own movement while receiving the leader mission signal. The leader robot reduces simultaneously its own moving speed to achieve the purpose of character switching. The experimental process is shown in Figure 24. Figure 25 shows the relative distance positions of the three robots. The initial moment, the initial coordinates of the starting point is $q_{l}=\left(600,520,60^{{}^{\circ}}\right)$, $q_{v}=\left(125,720,90^{{}^{\circ}}\right)$, $q_{f}=\left(125,260,90^{{}^{\circ}}\right)$, and the terminal coordinates is $q_{l}=\left(1900,710,45^{{}^{\circ}}\right)$, $q_{v}=\left(2190,620,20^{{}^{\circ}}\right)$, $q_{f}=\left(1820,210,10^{{}^{\circ}}\right)$, as shown in Figure 26. The three ASRs maintain a fixed distance of 245 mm and form the shape of an isosceles triangle. At a sampling point, Figure 27 shows that the relative distances of the $L_{lv}$, L and $L_{fv}$ are 100 mm, 150 mm and 340 mm respectively, because the leader robot sends the transfer command and reduces the speed of movement, the follower 1 robot increases the moving speed and pull away from the leader robot and closer to the follower 2 robot. The relative offset distance of the second sampling point is 200 mm, 150 mm and 600 mm respectively, at this time, the follower 1 robot has reached the leader position and the $L_{fv}$ distance reaches the maximum value. At the third sampling point, the follower 2 robot speeds up the movement and forms a formation again at the fourth sampling point, the experiment basically completes the purpose of the role interchange. The leader robot leading two follower robots in interchangeable relative positions and completing the formation algorithm at *t* = 4 s and complete the swapping position at the *t* = 8 s, the difference in speed between the three robots is shown in Figure 28. Thereby, the positional exchange of two follower robots is completed under the command of the leader robot.

### 4.4. Experiment IV: Visual Servoing Experiment

In order to verify the control effect of the visual servoing in the formation control processing, the underwater robot motion control experiment based on visual servoing module installed on a leader robot and a follower robot was performed, as shown in Figure 29, we designed a platform by using of the Openmv image sensors, and proposed two image recognition algorithms. The first is based on the method of the feature point recognition algorithm, and the other is the edge detection method based on threshold segmentation algorithm. It can be seen from Figure 30 that the method based on threshold analysis can detect spherical target objects more clearly and intuitively. (a), (b) and (c) in Figure 30 is based on the feature point recognition algorithm. It can be seen that Figure 30c fully realizes the purpose of clear edge feature point detection. Figure 30d–f is threshold detection methods based on the threshold segmentation algorithm. It can be seen from Figure 30f that the purpose of detecting the target is achieved, but the target image has a greater impact on external conditions (light, target distance, etc.) Therefore, for the target image recognition algorithm in a complex underwater environment, the algorithm based on edge feature point detection is more suitable for underwater target detection.

The main content of this experiment was to collect the fuzzy control method between the robot speed and the distance by using the PID algorithm, a follower robot tacks the lead robot using a visual servoing arithmetic, as shown in Figure 31, when the leader robot is found by the camera, the follower robot increases the moving speed and reaches the position of the leading robot, and then proceeds at the same speed under the action of the PID algorithm until the target position is reached. It can be seen from Figure 32 that there is a big difference in the speed control between the leader and follower robot without the PID algorithm and the PID algorithm. Without the PID control, the speed does not change after the maximum value at 6–8 s, the follower robot will continue to move after the leader robot and reach the terminal point. After approaching the target, the speed of the PID algorithm is close to zero and may cause the leader robot to crash and the speed may reach a maximum.

## 5. Conclusions

This paper proposed a novel formation strategy for ASRs to solve motion kinematic constraints of the multi-robot. The proposed control algorithms are based on stability and communication modules. The basic principle of achieving multiple robot collaboration, which used a follower robot provides the reference coordinates for leader robot and another follower robot, finally, the corresponding mathematical relationship formula is given. This paper also proposes a multi-robot change strategy based on the bidding control method, which can change the queue transformation in the case of the leader robot damage or cannot lead the follower robot to complete the purpose of the task. Moreover, this paper also proposes a method to determine the communication distance based on the delay characteristics of the communication module, the direct distance between robots is determined by the difference in the delay time when robots moving different displacement. More importantly, motion control strategies involving the visual servoing module, PID-based speed and distance control algorithm verify the effectiveness of the presented modeling and control method to realize multi robot collaboration and cooperation, it is found that the visual servoing is able to achieve continuous both propulsive speed and the ratio of thrust to control distance of two collaborative robots. Through the underwater experiments, the feasibility of the theory proposed by this paper is proved.

For future research, continuous improvement efforts on mechanical design and control approaches will be devoted to optimization of multi-robot cooperation. In addition, the theory of models can manage the dynamics system that multiple degrees of freedom to control multiple robots to achieve rounding and hunting is worthy of investigation.

## Figures and Tables

**Figure 1 micromachines-11-00071-f001:**
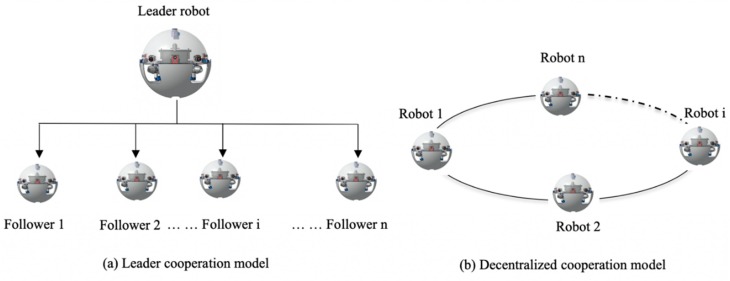
Schematic of the multi-robot cooperation (this study uses (**a**) mode: n = 3). (**a**) Leader cooperation model; (**b**) Decentralized cooperation model.

**Figure 2 micromachines-11-00071-f002:**
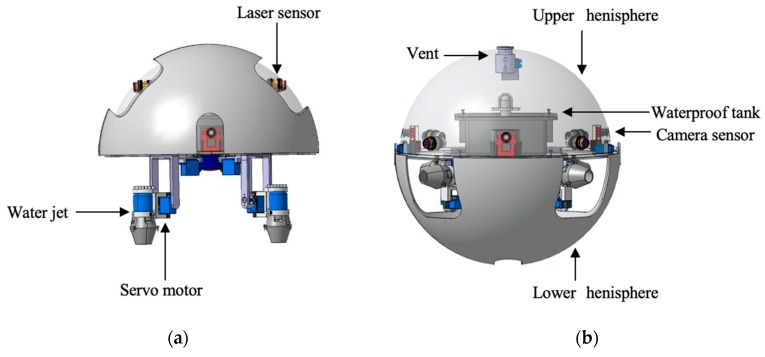
Two modes of the spherical amphibious robot. (**a**) underwater mode; (**b**) land mode.

**Figure 3 micromachines-11-00071-f003:**
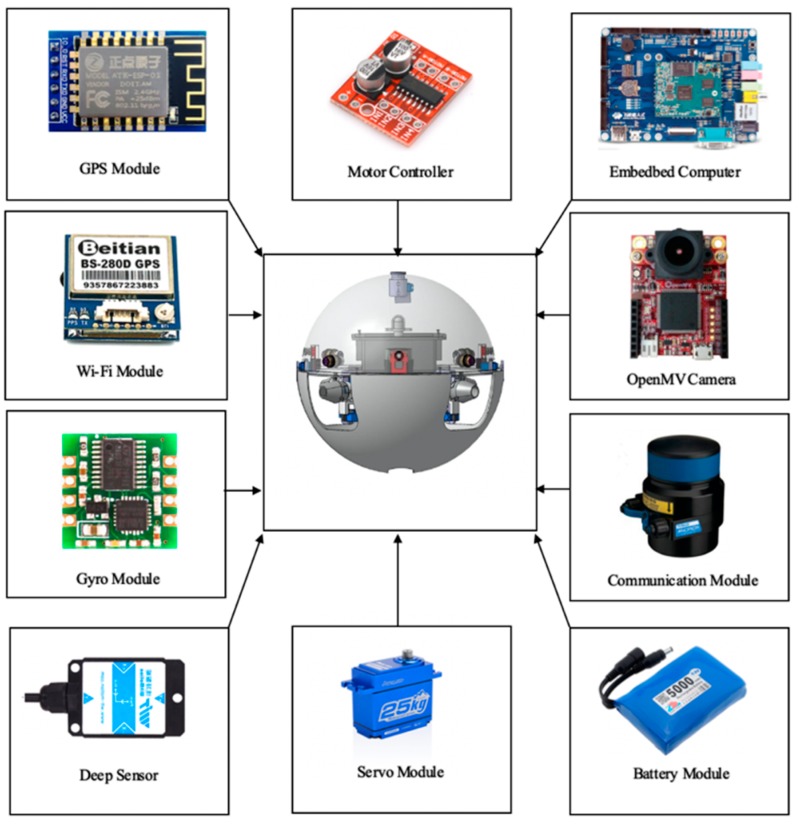
Composition of the electromagnetic actuation system.

**Figure 4 micromachines-11-00071-f004:**
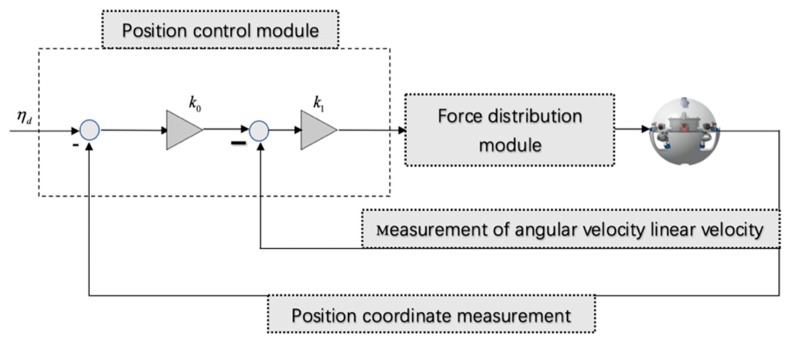
Aechitecture of the linear quadratic regulator control.

**Figure 5 micromachines-11-00071-f005:**
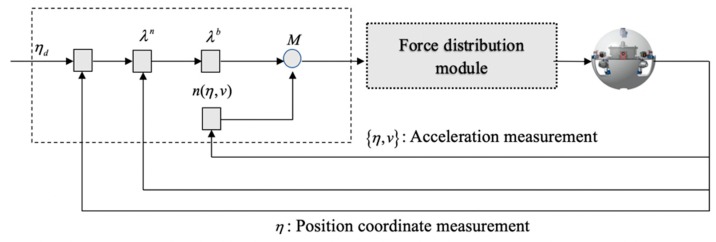
Aechitecture of the state feedback linearization controller.

**Figure 6 micromachines-11-00071-f006:**
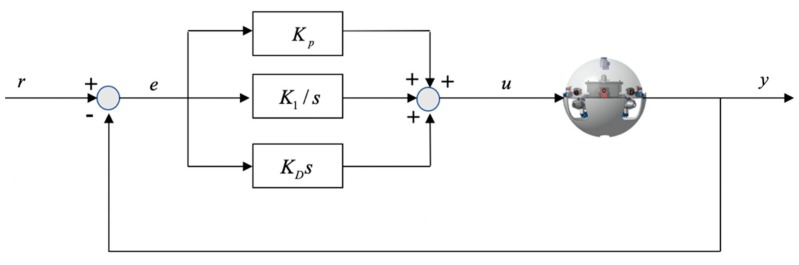
Architecture of the Proportion Integral Differential (PID) controller.

**Figure 7 micromachines-11-00071-f007:**
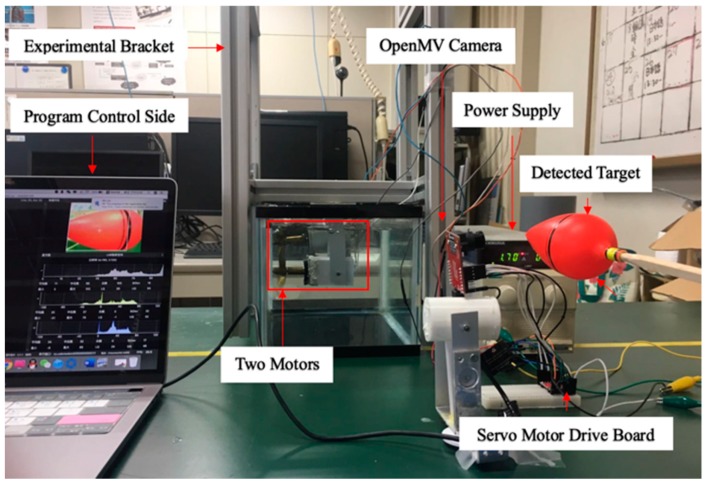
Experimental set up for a moving target identification.

**Figure 8 micromachines-11-00071-f008:**
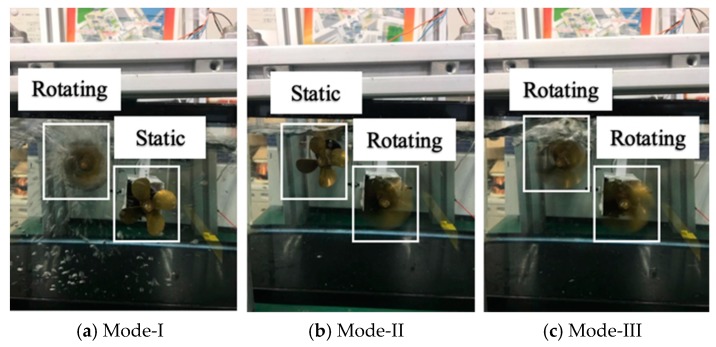
Two control modes of the visual servoing. (**a**,**b**) Looking for target mode; (**c**) Going forward for target mode.

**Figure 9 micromachines-11-00071-f009:**
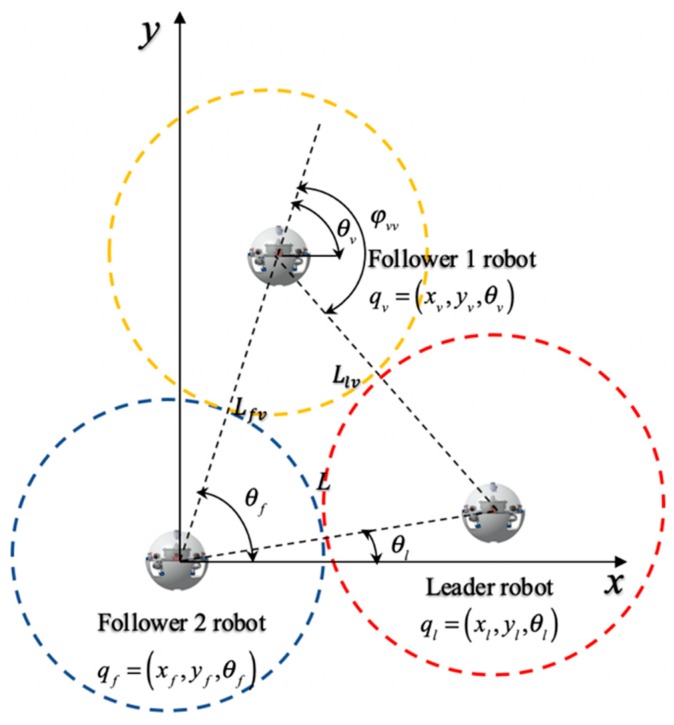
Multi-robot representation in the Cartesian coordinates.

**Figure 10 micromachines-11-00071-f010:**
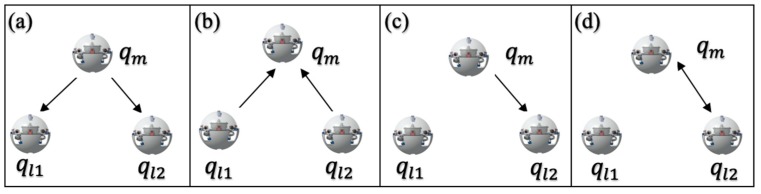
Contract net based on the task allocation. (**a**) Task transfer; (**b**) Bidding; (**c**) Authorization; (**d**) Confirmation.

**Figure 11 micromachines-11-00071-f011:**
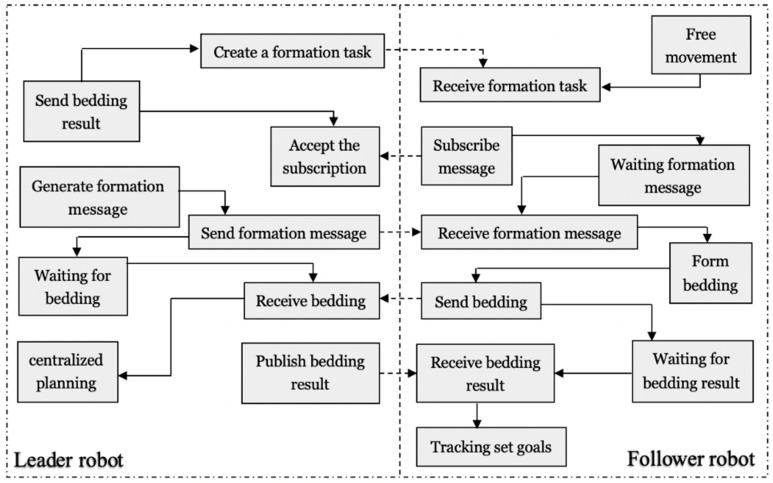
Flow diagram of the auction-based planning.

**Figure 12 micromachines-11-00071-f012:**
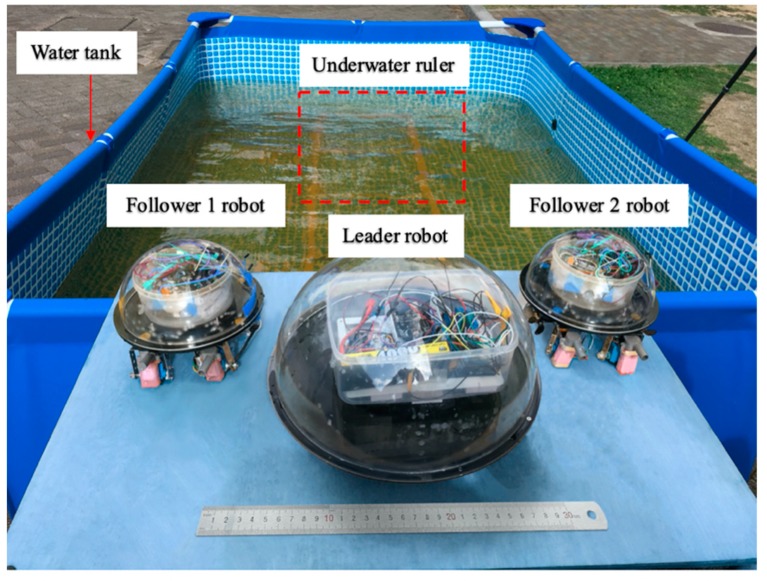
Experimental set up of the multi-robot cooperation.

**Figure 13 micromachines-11-00071-f013:**
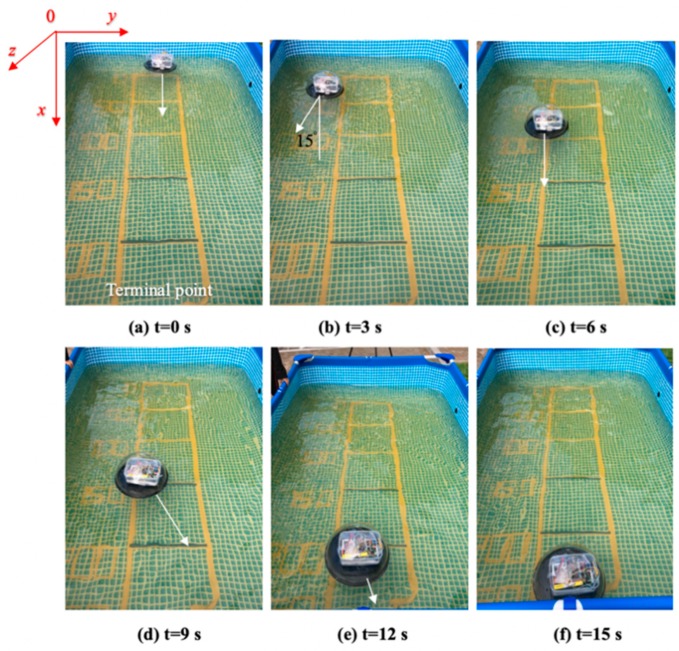
Stability control images of a single robot.

**Figure 14 micromachines-11-00071-f014:**
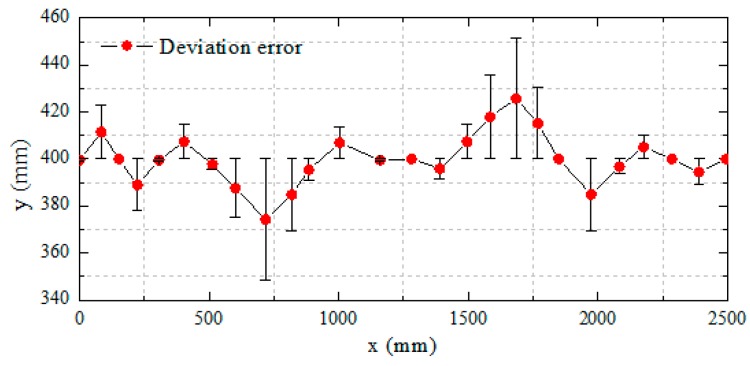
Deviation error curve in the y-axis direction.

**Figure 15 micromachines-11-00071-f015:**
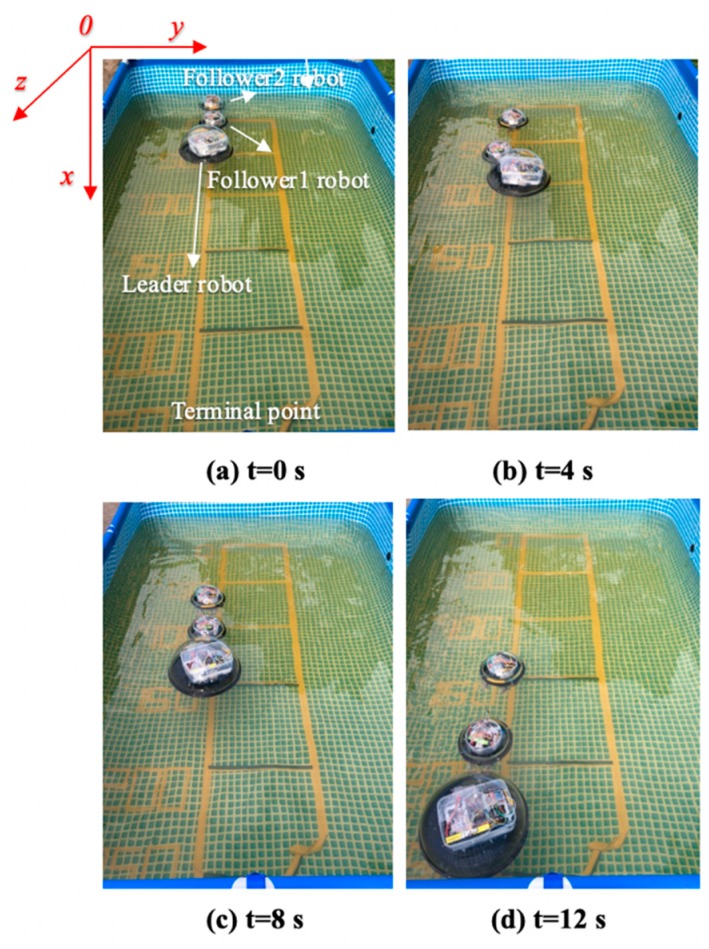
Relative distance between robots.

**Figure 16 micromachines-11-00071-f016:**
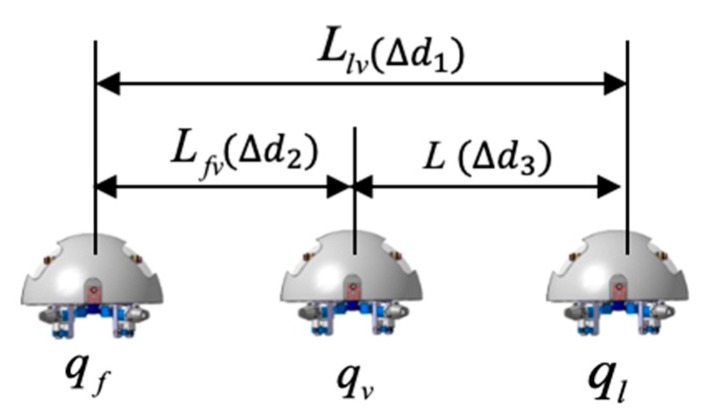
Robots formation experiment with a longitudinal queue.

**Figure 17 micromachines-11-00071-f017:**
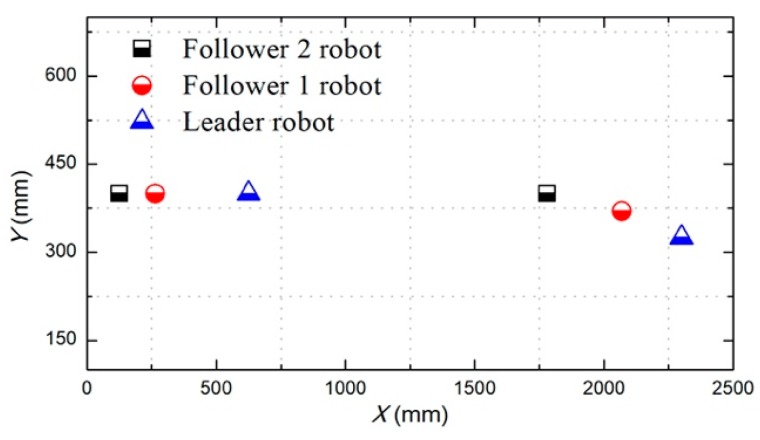
Calibration of the start and terminal coordinate (longitudinal queue).

**Figure 18 micromachines-11-00071-f018:**
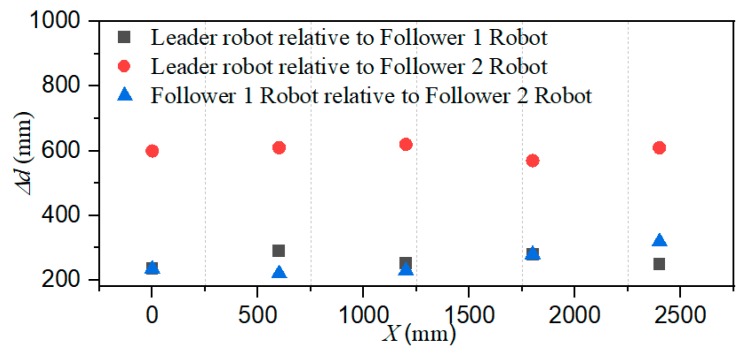
Relative localization offset with a longitudinal queue (Δd is the relative distance between two robots).

**Figure 19 micromachines-11-00071-f019:**
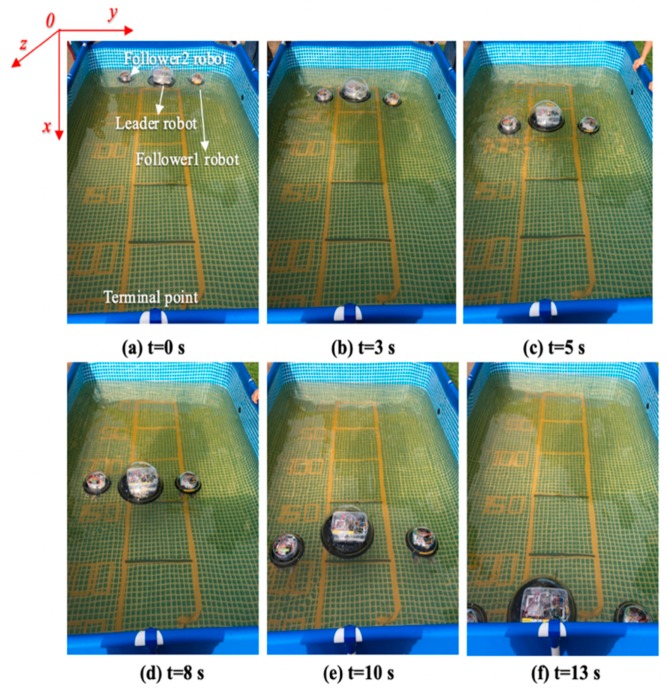
Robots formation experiment with a lateral queue.

**Figure 20 micromachines-11-00071-f020:**
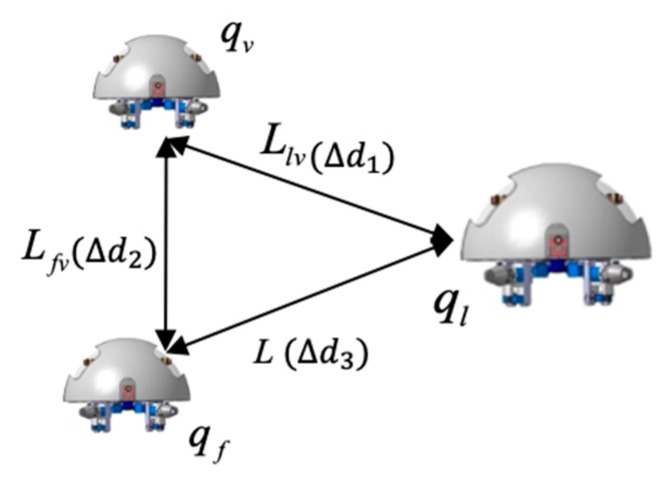
Relative distance between robots.

**Figure 21 micromachines-11-00071-f021:**
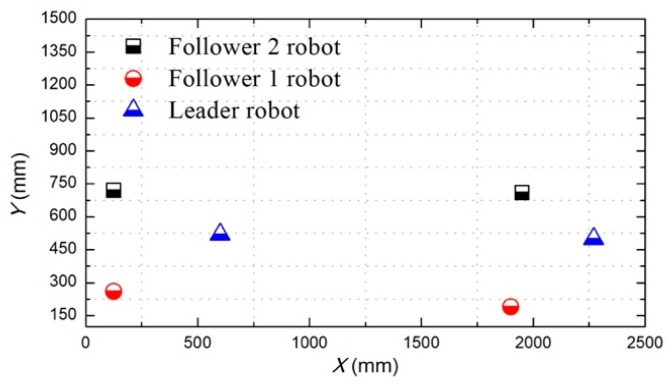
Calibration of the start and terminal coordinate (lateral queue).

**Figure 22 micromachines-11-00071-f022:**
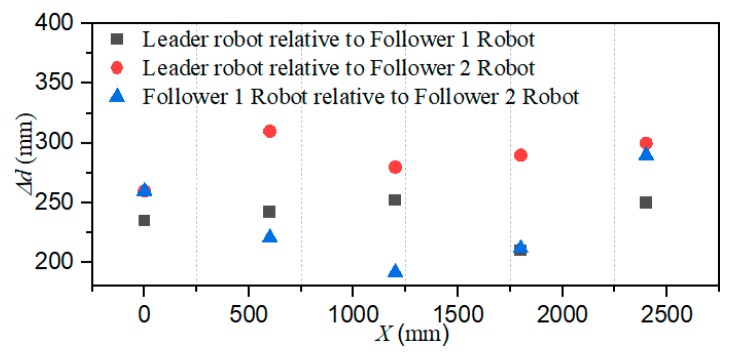
Relative localization offset with a lateral queue. (Δd is the relative distance between two robots).

**Figure 23 micromachines-11-00071-f023:**
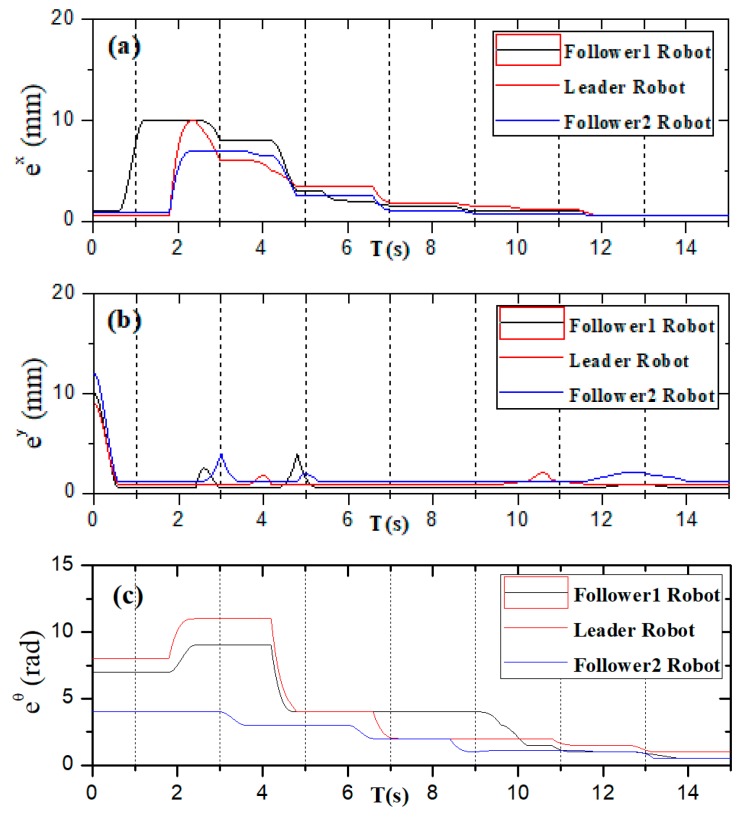
Error change of the x-axis, y-axis and rotation angle. (**a**) x-axis; (**b**) y-axis; (**c**) rotation angle.

**Figure 24 micromachines-11-00071-f024:**
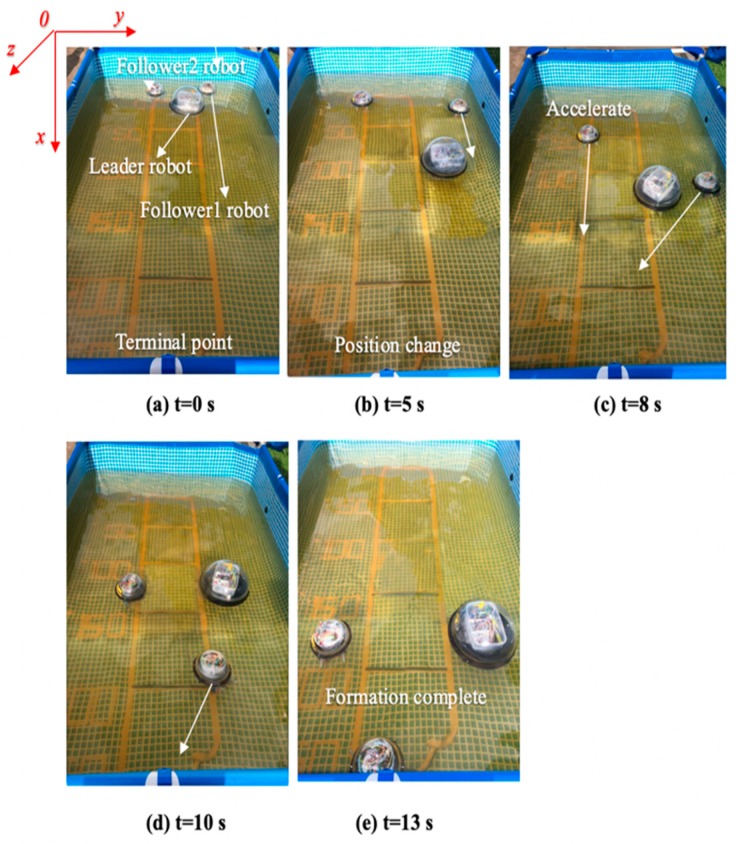
Experimental set up for a multi-robot role interchange.

**Figure 25 micromachines-11-00071-f025:**
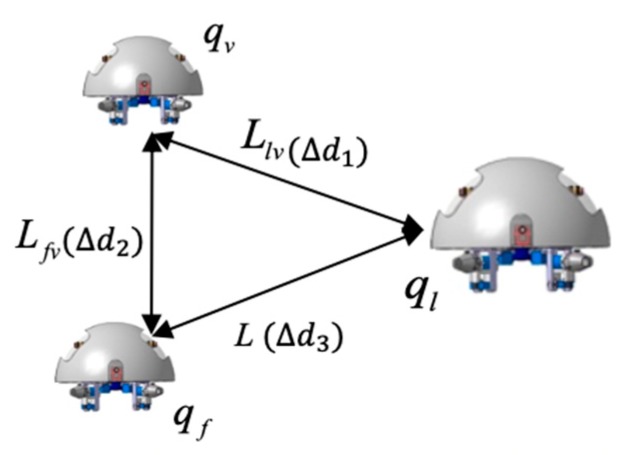
Relative distance between robots.

**Figure 26 micromachines-11-00071-f026:**
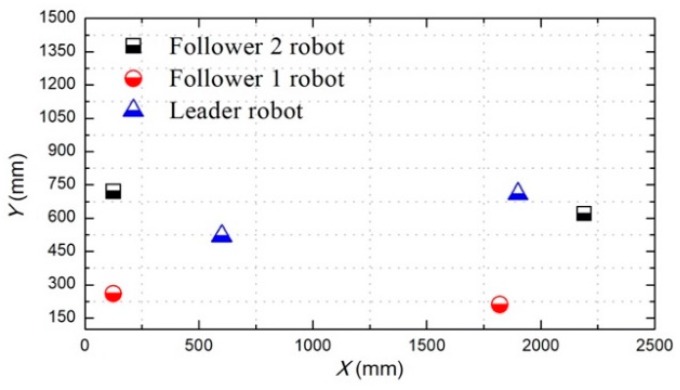
Calibration of the start and terminal coordinate (Interchange queue).

**Figure 27 micromachines-11-00071-f027:**
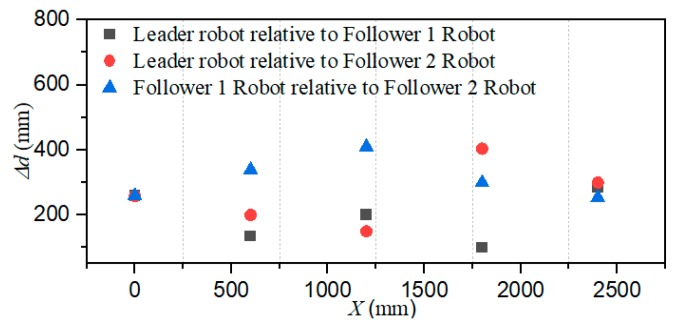
Relative localization offset with an interchange queue (Δd is the relative distance between two robots).

**Figure 28 micromachines-11-00071-f028:**
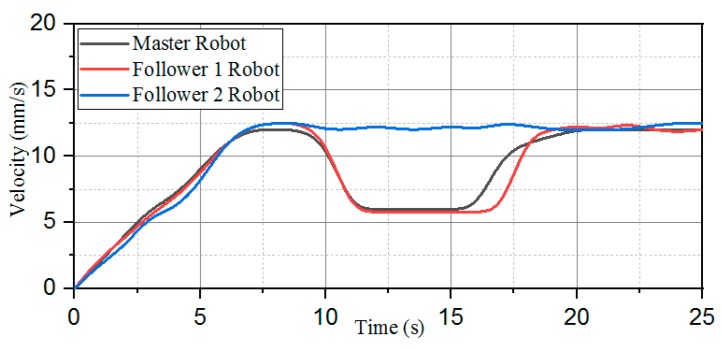
Velocity change of the robot character interchange.

**Figure 29 micromachines-11-00071-f029:**
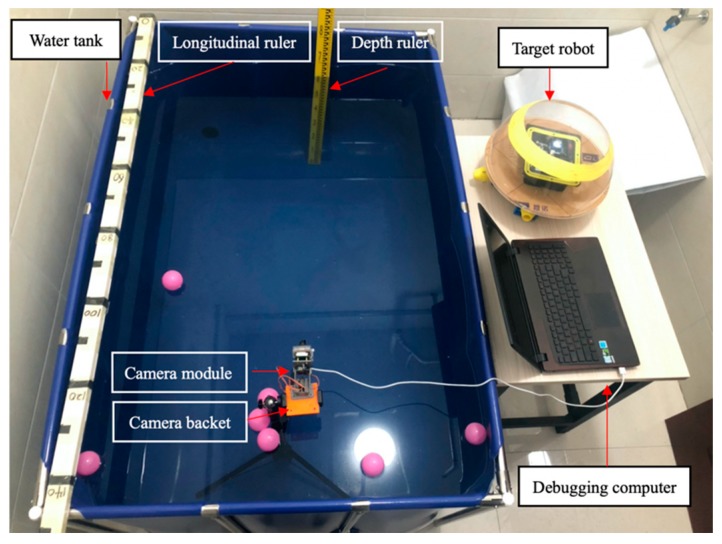
Experimental set up for thetarget image recognition.

**Figure 30 micromachines-11-00071-f030:**
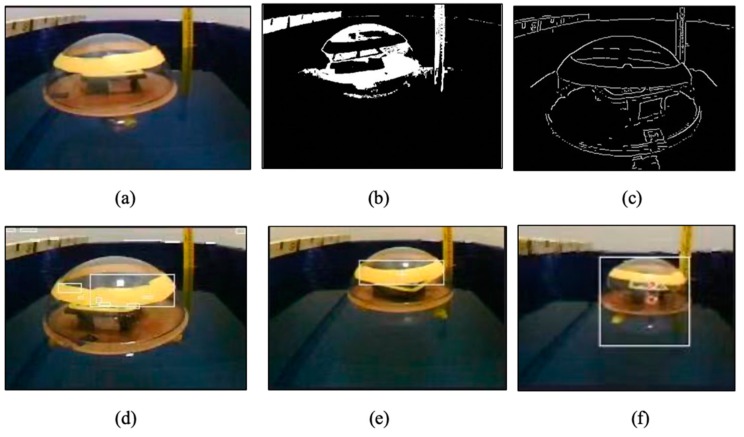
Comparison of two image recognition algorithms. (**a**–**c**) The feature point recognition algorithm; (**d**–**f**) The threshold segmentation algorithm.

**Figure 31 micromachines-11-00071-f031:**
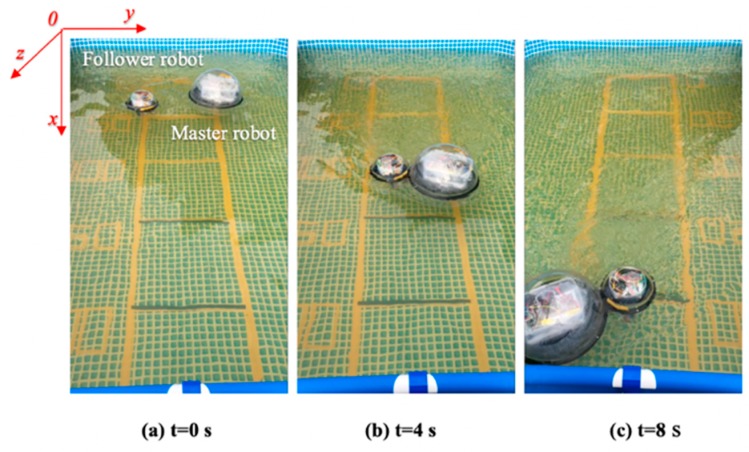
Following images based on the visual servoing algorithm.

**Figure 32 micromachines-11-00071-f032:**
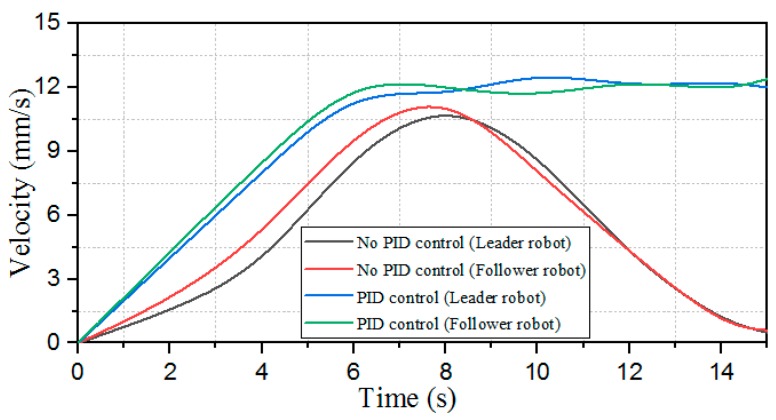
Experimental results of the velocity comparison.

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
