# Peer review of "Collaboration and Task Planning of Turtle-Inspired Multiple Amphibious Spherical Robots"

_micromachines, 2020, doi:10.3390/mi11010071_

Round 1

Reviewer 1 Report

I certainly miss the introduction to good works reported by authors like Rido or Ortiz. Please check, discuss and cite the most relevant. Justify the real contribution of your work in contrast to these and the rest of commented works in the manuscript.

https://cirs.udg.edu/

https://sites.google.com/site/pereridaorodriguez/publications

http://srv.uib.es/alberto-ortiz-rodriguez/

https://link.springer.com/chapter/10.1007%2F978-3-319-70833-1_4

Also, traditional tendencies in underwater environments have focused on sonar/acoustic field. Please also widen the discussion on this kind of communication-related aspect. The intro already covers in depth the kind of AUV robots availabe. More updates on the sort of wave-transmission technologies should be expected.

https://www.mdpi.com/2079-9292/8/11/1297

Visual servoing:

More details about the camera specs, image acquisition and preprocessing is needed. Please include some captions, and either more diagrams to clarify the procedure.

Why PID an not other control schemes? I know that dispite being one of the simplies and older control schemes, in some cases performs sufficiently. Nontheless, this discussion has to be addressed.

Authors claim to compare but they only enable/disable the PID control. The experimental setup needs to be improve with more formal and valid comparisons, ie: the sort of control, and image processing such as image filtering, resolution variation, bandwith allocation, lighting conditions...

My first main concern regards to this last point.

My second, and even worst concer goes to the relation to authors' references [31-38] in section 2.1. I assume and understand that this is a long term project with sequential improvements and novelties. However, if authors describe the whole design based on previous works, the introduction of newly contributions and novelties should be clearly devised and explained in depth.

I intuit that such contributions with respect to latest authors' papers published in 2019 with this robot, regards to the visual servoing. That leads us to my first main concern.

Author Response

Dear Reviewers,

I am very grateful to your comments for the manuscript. All your suggestions are very important, and they have the important guiding significance on my thesis writing and research work. The list of responses to the comments is followed:

Responses to Reviewers

To Reviewer #1:

I certainly miss the introduction to good works reported by authors like Rido or Ortiz. Please check, discuss and cite the most relevant. Justify the real contribution of your work in contrast to these and the rest of commented works in the manuscript.

Response: Thank you for your comments. I have carefully reviewed the papers and reference. I found that my paper existed many problems. I have made strict changes to the order of the reference, and carefully read the relevant research papers you recommend and quote some of important papers. They have been marked with red font in the section I of the revised manuscript.

Also, traditional tendencies in underwater environments have focused on sonar/acoustic field. Please also widen the discussion on this kind of communication-related aspect. The intro already covers in depth the kind of AUV robots available. More updates on the sort of wave-transmission technologies should be expected.

Response: Thank you for your comments. The detailed communication principle has been given in section II, and the discussion also has been given. Due to the experimental limitations of the underwater communication system, the research in this paper only tests the underwater sonar module. In the future research, we will compare various underwater communication methods and take more related experiments, Thanks for your understanding.

More details about the camera specs, image acquisition and preprocessing are needed. Please include some captions, and either more diagrams to clarify the procedure.

Response: Thank you for your suggestions. Your comments are very meaningful for this study. In Section 4.4, two algorithms for the camera image recognition were redesigned, as shown in Figure 29 and Figure 30. The first picture mainly describes the experimental environment. The second picture is an image test of the two algorithms. At last, we draw conclusions based on impact and effectiveness.

Why PID is not used other control schemes? I know that despite being one of the simplies and older control schemes, in some cases performs sufficiently. nonetheless, this discussion has to be addressed.

Response: Thank you for your suggestions. Your opinion is exactly what I am worried about, and I have carefully revised it according to your opinions. I made some corrects according to your comments. Three models of controllers and comparison of advantages and disadvantages have been written in section II. We are also added models and mathematical expressions for the three controllers,as shown in Figure 4, 5, and 6. The principles of LQR controller, FL controller and PID controller have been compared in detail (In section 2.3), and the effectiveness and simplicity of the simple PID algorithms for motion control in complex underwater environments are given in this section. They all have been marked with red font in the revised manuscript.  

My second, and even worst concern goes to the relation to authors' references [31-38] in section 2.1. I assume and understand that this is a long-term project with sequential improvements and novelties. However, if authors describe the whole design based on previous works, the introduction of newly contributions and novelties should be clearly devised and explained in depth.

Response: Thank you for your suggestions. The references [31-38] has been changed to [36-42]. At the end of the paragraph, the connection and difference between the new ASR proposed by this paper and the previous design are increased, and the foremost innovation points of the article are given. They have been written on page four and marked with red font in the revised manuscript.

I intuit that such contributions with respect to latest authors' papers published in 2019 with this robot, regards to the visual servoing. That leads us to my first main concern.

Response: Thank you for your advices. I read some related papers nearby 10 days on visual servoing published in 2019 and selected three valuable articles [33-35] for citation. In the future writing, I will try to quote the literature references of the past 2 years.

Reviewer 2 Report

In Abstract, the authors mention four categories of their objective. They have to clarify how the number four is counted. "Sover motor" in Figure 2 could be a wrong expression. Whenever any variable or coordinate first time appears in a manuscript, authors have to define it. Unfortunately, for instance, coordinates xv, yv, and thetav in Equation (2) are not defined. Furthermore, it is not reasonable that Equation (2) follows from Equation (1). More than once, "W" in where can not be capital. Styles of "X (mm)" in Figures 14 and 15 are not consistent. Below Figure 15, "L" and "Lfv" in "In Figure 19, the L and Lfv..........." are not placed on the same height. This lousy writing appears many times in the manuscript.

I think Algorithms 1 and 2 in Appendix are not required and should be deleted to save space. The authors should, however, append the proof of control stability to convince readers that theoretically their visual control method for amphibious robots is stable during control. In addition to Figure 8, a flow diagram, the authors have to append control block diagrams, so that readers can realize how visual servo is achieved and PID control and fuzzy control, mentioned in Experiment 4, are integrated.

In Figure 27, Velocity(cm) is a wrong expression. The authors have to give the hardware information; e.g. gyro sensors, camera, image processing card, interface card. In Author Contributions, electromagnetic actuation systems are mentioned. The manuscript has to describe the electromagnetic actuation systems including photos.

Author Response

Dear Reviewers,

I am very grateful to your comments for the manuscript. All your suggestions are very important, and they have the important guiding significance on my thesis writing and research work. The list of responses to the comments is followed:

Responses to Reviewers

To Reviewer #2:

In Abstract, the authors mention four categories of their objective. They have to clarify how the number four is counted. "Sover motor" in Figure 2 could be a wrong expression.

Response: Thank you for your comments. The four points in this paper are mainly designed from application value and research level. I am very sorry that the description of the servoing motor in the picture is wrong, I have made corrections in Figure.2 and replaced the structure of the robot with a novel structural design (contained some sensor designs), as shown in Figure 2, thank you for your reminder.

Whenever any variable or coordinate first time appears in a manuscript, authors have to define it. Unfortunately, for instance, coordinates xv, yv, and thetav in Equation (2) are not defined. Furthermore, it is not reasonable that Equation (2) follows from Equation (1). More than once, "W" in where can not be capital. Styles of "X (mm)" in Figures 14 and 15 are not consistent. Below Figure 15, "L" and "Lfv" in "In Figure 19, the L and Lfv..........." are not placed on the same height. This lousy writing appears many times in the manuscript.

Response: Thank you for your advices, I have to apologize deeply for this serious mistake. I carefully modified the origin of the formula and made detailed comments on the parameters inside the formula, such as formulas (9)-(14). The width and height of some formulas are not constant, and I have also made serious modifications. For example, on pages 13-17 of this paper, the modified parts are marked with red font.

I think Algorithms 1 and 2 in Appendix are not required and should be deleted to save space. The authors should, however, append the proof of control stability to convince readers that theoretically their visual control method for amphibious robots is stable during control. In addition to Figure 8, a flow diagram, the authors have to append control block diagrams, so that readers can realize how visual servo is achieved and PID control and fuzzy control, mentioned in Experiment 4, are integrated.

Response: Thank you for your suggestions. Algorithms 1 and 2 have been removed from this paper to save space, and I also added models and mathematical expressions for the three controllers,as shown in Figure 4, 5, and 6. The principles of LQR controller, FL controller and PID controller have been compared in detail (In section 2.3), and the effectiveness and simplicity of simple PID algorithms for motion control in complex underwater environments are proposed in the section 2.3.

In Figure 27, Velocity(cm) is a wrong expression. The authors have to give the hardware information; e.g. gyro sensors, camera, image processing card, interface card. In Author Contributions, electromagnetic actuation systems are mentioned. The manuscript has to describe the electromagnetic actuation systems including photos.

Response: Thank you for your suggestions. Figure 27 has been modified to express the correct meaning with the Figure 32. To get better describe the design principles of electronic systems, I modified the Figure 2 with structure of the electromagnetic actuation systems, and marked with red font, the resolution of Figures 2 and 3 has been improved in the revised manuscript.

Round 2

Reviewer 1 Report

I appreciate that authors have sufficiently improved this paper based on my previous comments. However, I encourage them to keep in mind some uncompletely covered comments for their next future work.

Reviewer 2 Report

Concerning writing in this manuscript, any of "where...." has to be moved to the leftmost position. In addition, any of of "the Eq.(" has to be corrected to become "Eq.(" Concerning velocity units in Figure 32, it can be Velocity (cm/s) or Velocity (mm/s), but can not be Velocity (cm) or Velocity (mm). cm and mm are displacement units, not velocity units.